# Animal Welfare Problems in Sheep Farming: A Current Overview for Germany Based on Surveys of Veterinary Offices and the Evaluation of Publicly Accessible Court Cases

**DOI:** 10.3390/ani15142116

**Published:** 2025-07-17

**Authors:** Svenja Niethammer, Sarah Schmid, Hannah Hümmelchen, Axel Wehrend, Henrik Wagner

**Affiliations:** Veterinary Clinic for Reproductive Medicine and Neonatology, Justus Liebig University, 35392 Giessen, Germany; svenja.niethammer@uni-giessen.de (S.N.); sarah.schmid-2@vetmed.uni-giessen.de (S.S.); hannah.huemmelchen@vetmed.uni-giessen.de (H.H.); axel.wehrend@vetmed.uni-giessen.de (A.W.)

**Keywords:** animal welfare, sheep farming, veterinary, small ruminants

## Abstract

Animal welfare in sheep farming remains a relevant issue. To help prevent incidents of animal welfare violations, a nationwide analysis was conducted for the first time to identify the areas in which animal welfare problems occur most frequently in sheep farming in Germany. Until now, such data collection has been limited to isolated surveys conducted on individual farms. To this end, a questionnaire was sent to all veterinary authorities in Germany and online court decisions were evaluated. These two approaches allow for the cost-effective identification of animal welfare issues on a supra-regional level, thereby enabling the development of targeted improvements and preventive measures. Moreover, this methodology provides a basis for assessing potential changes in the status quo over time.

## 1. Introduction

Although sheep are considered a minor species in Germany with around 1.5 million animals and declining numbers [1], they are very important in Europe. Europe has the greatest diversity of breeds, with 60 million sheep [2] and 770 sheep breeds, accounting for half of all known sheep breeds worldwide [3]. Compared with other livestock species, sheep farming is less criticized by society [3,4]. This is because sheep are usually only kept indoors during the winter and spend the rest of the year on pastures or under herding systems, as landscape maintenance is an important economic sector within sheep farming [4]. However, sheep farmers can come under public scrutiny, depending on the type of farming, as the animals are often visible to the public, such as during dike shepherding or in transhumance sheep farms. If a member of the public reports a suspected animal welfare violation in Germany, the authorized veterinary office is obliged to conduct an animal welfare inspection of the farm, as veterinary offices are the first administrative authorities responsible for implementing the Animal Welfare Act. Therefore, they play a central role in monitoring and evaluating animal welfare in sheep farms. If a conflict cannot be resolved, the case is referred to a court. These cases are publicly accessible in Germany.

Currently, the general provisions of the Animal Welfare Livestock Farming Ordinance and the Animal Welfare Act, which came into force in 1972, also apply to sheep farming in Germany. Since 2002, animal welfare has been enshrined as a state objective in the German Constitution [5].

To improve animal welfare, prevention of animal suffering and disease is a fundamental aspect of animal protection [6]. One of the most important factors influencing animal welfare is animal-friendly housing [7], which should be evaluated daily by the livestock owner, along with the animal’s state of health, to ensure a safe and hygienic environment for the animals [8]. Effective parasite management is also required in addition to good hygiene management. This is particularly important for long-tailed sheep breeds [9], as fecal contamination on the tail and anogenital area can lead to further problems, such as fly maggot infestation [10]. Therefore, several sheep farmers dock the tails of their animals [11]; however, according to the German Animal Welfare Act, this is only permitted in exceptional cases, such as when it is needed for the protection of the animal [12]. However, according to the draft law, this exception will be completely banned after a transitional period of 8 years [13].

Knowledge of the most common animal welfare problems in sheep farming is important for introducing and implementing targeted preventive measures. Although these problems have been recognized for some time [14,15], to date, there has been no comprehensive, cross-regional data analysis on current animal welfare problems in sheep in Germany.

## 2. Material and Methods

As part of the project “Practical Implementation and Improvement of Animal Welfare in German Sheep Farming Using the Example of Undocked Animals” (“Animal Welfare Competence Centre Sheep” in short) funded by the Federal Agency for Agriculture and Food, a questionnaire aimed at encouraging all German veterinary offices to record the situation of animal welfare cases and problems in German sheep farming was developed. The questionnaire consisted of 26 questions, which allow multiple and free-text answers. The questionnaire was completed anonymously online from 9 August 2023 to 30 September 2023. All German veterinary offices (393) were contacted by email, which they could use to access the online questionnaire directly. Among other things, they were asked regarding the types of animal welfare violations they have reported or found. The question was divided into six subcategories in which the participants were able to indicate whether these problems had occurred by providing multiple answers. Importantly, in Germany, animal welfare violations can be reported by anyone—including private individuals, neighbors, hikers, or passersby—not only by professionals such as veterinarians or animal welfare officers.

Additionally, data on animal welfare cases in sheep farming over the last 30 years, which were accessible online and heard by German courts, were collected. The violations dealt with were divided into the categories “husbandry and housing”, “feed and water supply”, “health”, “lambing”, “management”, and “other”, with several categories possible per case. The questionnaire is provided in the Appendix A. The website https://openjur.de/ was used to conduct research on publicly accessible court cases (accessed in 2023).

## 3. Results

### 3.1. Evaluation of the Survey of the Veterinary Offices

A total of 71 (approximately 18.1%) of the 393 contacted veterinary offices participated in the questionnaire survey. Most participating offices were from the federal states of North Rhine-Westphalia and Baden-Württemberg (Table 1). The highest number of participating offices in relation to the number of veterinary offices in each federal state was found in the federal states of Brandenburg and Thuringia.

The survey revealed that an average of 5907.1 sheep (range, 15–31,503; median, 3814) were registered on 357.2 farms (range, 3–2424; median, 265) in the areas of responsibility of the participating veterinary offices. Most sheep farms and sheep were registered in the federal states of Mecklenburg-Western Pomerania, Saxony, and Saxony-Anhalt.

Sixty-seven veterinary offices answered questions regarding the size of the farms in the respective areas of responsibility. Of these, 66 offices stated that they were responsible for holdings with 1–10 sheep. Farm sizes of 11–50 animals were represented at 54 offices. The number of veterinary offices responsible for farms with >50 animals fell sharply compared with smaller farms. Holdings of 51–100 sheep were looked after by six veterinary offices. Farms with 101–500 and 501–1000 animals were each looked after by two veterinary offices, and only one veterinary office was responsible for farms with more than 1000 animals.

A total of 42 veterinary offices answered the question regarding the average number of reported animal welfare cases per year within the last five years. The percentage distribution was as follows: 5–10 cases per year, 45.2% (n = 19); 10–15 cases per year, 23.8% (n = 10); 15–20 cases per year, 7.1% (n = 3); and >20 cases per year, 23.8% (n = 10).

#### 3.1.1. Animal Welfare Violations by Category

a.Husbandry and housing

Violations related to weather protection were most frequently mentioned (Table 2). No differences were found in the geographical distribution. Other problems in the area of husbandry and housing included, for example, the risk of injury to the animals due to housing or individual housing.

b.Feed and water supply

The most frequent animal welfare cases were observed in the water supply to the animals (Table 3). In addition, no differences in geographical distribution were observed.

c.Health

The most common health issues were lameness, downer sheep, and animals with skin damage (Table 4).

d.Lambing

Increased lamb mortality was most frequently reported (Table 5). Other problems related to lambing included lambing outdoors in bad weather (n = 6) and animals being left unnoticed in the lambing area by transhumant shepherds (n = 1).

e.Management

Shearing represented a potential conflict for more than three-quarters of the participating offices (Table 6). Problems with livestock-guarding dogs included risks to walkers (n = 2). Other problems mentioned included an increased risk of injury, lack of expertise, keeping animals alone, and poor management by transhumance shepherds who failed to notice sick and injured animals in a timely manner.

f.Other

In addition to the categories already mentioned, 11 participating offices mentioned other issues, such as poor hygiene (n = 1), animal welfare during slaughter (n = 2), and disposal of dead animals (n = 5).

#### 3.1.2. Influence of the Seasons on the Frequency of Animal Welfare Cases

A total of 65 veterinary offices provided information on the seasonal accumulation of animal welfare cases. Of these, 43.1% (n = 28) reported that most animal welfare complaints occurred during the summer months (June to August). This was the most frequently mentioned response option in almost all the federal states. In Berlin, Lower Saxony, and Rhineland-Palatinate, the most frequent response was that the season had no influence, whereas in Thuringia, the lack of influence of the season was stated to be as frequent (40%) as the increased occurrence of animal welfare cases during the summer months. Of the 65 offices, 21 (32.3%) found that the season had no influence. Whereas, 11 (16.9%) offices reported seasonal accumulation in the winter months (December to February) followed in third place. Clusters were rarely observed in spring (March to May, 6.2%) or fall (September to November, 1.5%). A cluster occurred in the spring in the federal states of Hesse, Lower Saxony, Saxony, and Schleswig-Holstein. In North Rhine-Westphalia, a veterinary office recorded a cluster during winter.

#### 3.1.3. Follow-Up and Progress of Animal Welfare Cases

Misjudgments of the situation by the reporter occurred in 26–50% of animal welfare complaints in 21 of the 60 offices that answered this sub-question. Eleven participating offices stated that 51–75% of the reports they received were the result of an incorrect assessment of the situation. Eight offices stated that misjudgments occurred in 6–10% or 16–20% of cases. Five participating offices reported a frequency of 21–25%, whereas four and two offices reported frequencies of 76–100% and 11–15%, respectively. One veterinary office consistently provided accurate assessments.

Most animal welfare cases in sheep farms were settled out of court. Only in one veterinary office did animal welfare cases in sheep farming usually end in court proceedings (Table 7).

Most veterinary offices stated that animal removal, keeping, and care bans had not yet been implemented. Overall, the figures were similar for all three measures. Only in the area of animal removal did three offices state that animal removal was conducted in 51–75% of the cases. In the area of animal keeping and care bans, the frequency was at most 26–50% (Table 8).

Forty-three veterinary offices answered the question regarding the frequency of compliance-relevant infringements. Of these, 39.5% stated no cross-compliance-relevant infringements during conditionality checks. Furthermore, 14% stated that there were infringements in 51–75% of the checks conducted. Violations were found in 6–10%, 16–20%, 21–25%, or 26–50% of the checks in 9.3% (n = 4) each. Three participating offices stated that infringements were found in 11–15% of the checks, and one office reported infringements in 76–100% of the checks. Where infringements were found, the most frequent problems were in animal identification or maintaining the herd register. Weather protection, veterinary care, and water supply for animals were also cited more frequently. The prescribed identification of the animals was conducted correctly, either 51–75% or 76–100% of the time (42.6% each). Approximately 8.2% stated that the animals were correctly identified 26–50% of the time, and 4.9% stated that this was the case 21–25% of the time. Only one participating office stated that animals in their area of responsibility were never properly labeled.

Repeat cases of non-compliance occurred in at least every second farm in 26% of the veterinary offices that answered this question. Only one participating office never had any repeat cases (Figure 1).

In 23.3% of the 60 veterinary offices that provided information, more than 76% of animal owners were very compliant when informed of animal welfare violations on the farm. Approximately 35% stated that owner compliance was optimal in 51–75% of cases. Only two veterinary offices had a compliance rate of <10% (Figure 2).

#### 3.1.4. Tail Docking

In farms reported due to an animal welfare violations, the sheep were mostly docked in 10 (16.4%) of the 61 respondents. Meanwhile, the animals were mostly undocked in 36.1% of the participating offices who answered this question. A total of 29 participating offices (47.5%) were unable to answer this question.

Fifty veterinary offices responded to the question of whether increased problems had been identified in sheep with undocked tails. Of these, 92% of the participating offices (n = 46) found no increased incidence of fly maggot infestation or diarrhea in undocked animals compared with docked animals. Only 2 (4%) of these animals had diarrhea or fly maggots.

#### 3.1.5. Killings

Improper killing of sheep was noted by two veterinary offices in 11–15% and 16–20% of the cases, respectively. Three participating offices observed it in 6–10% of the cases. In 82.5% of the 40 respondents who answered this sub-question, sheep were never killed improperly. Notably, the problems encountered were mainly secondary to a lack of knowledge or skills (Table 9).

#### 3.1.6. Animal Transportation

A total of 36 veterinary offices provided information on animal welfare problems that occurred during transportation. The following answer options were provided in the area of transportability: sheep that were unable to walk (38.9%), female sheep in the last tenth of pregnancy or up to 7 days after lambing (22.2%), lambs younger than 1 week (22.2%), and sheep with large wounds (16.7%). The most common criticism of the transporter was that the stocking density was too high (38.9%), followed by the risk of injury to animals (27.8%), deficiencies in ventilation (22.2%), insufficient floor stability (19.4%), and the risk of sheep falling off the trailer during transportation (19.4%). Other deficiencies that were mentioned were an unsuitable transport vehicle (n = 2) and an excessively long loading time (n = 1).

The number of animal welfare violations in the last 5 years remained almost the same in 75% of the offices. Fourteen participating offices (21.9%) stated that the number of animal welfare violations had increased, whereas 3.1% (n = 2) stated that the number had decreased.

At the end of the questionnaire, the participants had the opportunity to make their own comment on animal welfare in sheep farming. The most common comment (n = 6) was that hobbies involving little expertise are problematic. Two participating offices also noted that a lack of expertise could lead to animal welfare problems. The fact that a veterinarian is often consulted too late or not at all (n = 3) was also criticized. One participating office stated that good parasite monitoring was often lacking. The availability of sheep shearers (n = 3) was also a problem. Two participating offices mentioned problems with the illegal slaughter of sheep or the keeping of herding dogs. Two comments addressed the issue of tail docking in sheep. Almost all sheep were docked, although the length of the tail had increased over the years. The ban on docking was poorly enforced by authorities. Castration of ram lambs (n = 1) and lamb loss due to ravens (n = 1) were also mentioned.

### 3.2. Evaluation of Publicly Accessible Court Cases

The evaluation of court rulings included 52 cases from 11 federal states. On average, 2.75 ± 1.5 (range, 1–7) violations were identified per case. Most of the data were generated in Bavaria, followed by Lower Saxony, North Rhine-Westphalia, and Rhineland-Palatinate (Table 10).

In half of the cases, sheep had inadequate nutrition (Table 11). More frequent problems included inadequate water supply, insufficient protection from weather, and incorrect shearing management. Other reasons included keeping animals despite the ban on keeping animals or a rope tied around the sheep’s neck to make it easier to catch them. In approximately one-fifth of cases, animal care and parasite management were inadequate. Other problems included lack of treatment, inadequate fencing or stabling, lameness, inadequate animal care, and lack of identification.

The most common measure was a ban on the maintenance of animals (n = 22). Animal removal (n = 9), herd reduction (n = 3), and herd dissolution (n = 4) were also reported. Correction of infringements was requested in six cases. A fine had to be paid in five cases, and a prison sentence was imposed in two cases. No consequences were observed in five cases.

## 4. Discussion

Improving animal welfare in sheep farming is an important social task [16,17]. One method involves presenting the current situation to identify and counteract existing deficiencies.

This data analysis represents the first Germany-wide collection of animal welfare cases and court rulings on sheep farming. No such data collection in other countries was found. To date, data on animal welfare problems in sheep have been collected from farms during transportation and slaughter [18,19,20]. Moreover, the types of sheep farming and animal welfare legislation vary greatly among countries, making it inappropriate to directly apply findings from other countries, such as Australian studies [20], to Germany. The approach of obtaining information relevant to animal welfare from the data of monitoring authorities and courts is new. However, this approach resulted in recording only selected cases. The advantage of this method is that data that are already available can be analyzed, ensuring that no costs are incurred for stock visits [21]. Therefore, the cross-regional analysis of animal welfare cases from the information sources presented could help obtain a better overall picture of the animal welfare situation and develop preventive approaches from the weaknesses uncovered in the future.

The participation of veterinary authorities in the survey was considered low, but it was within the expected range of other surveys [22,23]. Consequently, the evaluation only reflects a part of German sheep farming, and the number of participants varies within individual federal states. Therefore, although the results are not representative of the situation in Germany, they provide information on the areas in which animal welfare violations occur most frequently.

All the respondents stated that they handled at least five animal welfare cases annually. The tendency of the authorities to handle approximately the same number of animal welfare cases is striking. Therefore, animal welfare cases involving sheep are an important topic and field of activity for monitoring authorities. This underscores the need to educate sheep farms to avoid situations that are contrary to animal welfare. This transfer of knowledge could take the form of training courses for livestock owners. Studies have shown that training in animal handling leads to better handling of animals by livestock owners and, therefore, has a positive impact on animal behavior and welfare [24,25]. Such required knowledge and skills of animal owners or caregivers are among the principles of animal welfare [6].

Knowledge of the most common animal welfare problems is required to provide targeted information to sheep farmers and to prevent animal welfare problems. Both the survey of the veterinary offices and the online research on animal welfare cases in sheep farming revealed similar focal points for violations. These were mainly problems with adequate weather protection for the animals, nutritional deficiencies, a lack of water supply, and animals that were shorn too late or not at all.

The frequent occurrence of insufficient water supply may stem from the widespread belief among shepherds that animals grazing on fresh pasture do not require an additional water source. While sheep generally consume little extra water when fed on high-moisture forage, a water supply should still be provided at all times, as their water needs can increase significantly depending on performance level and ambient temperature. Here, better education of livestock owners is needed. The German Animal Welfare Livestock Farming Ordinance clearly requires that farm animals have constant access to water in sufficient quantity and quality [26]. According to the recommendation of the German Small Ruminant Veterinary Association, watering one to two times daily is considered sufficient, as the rumen is able to store large quantities of water [27].

The definition of “adequate weather protection” remains vague in the German Animal Welfare Livestock Farming Ordinance and is likely interpreted differently by individual veterinary offices. It states that animals must be provided “sufficient protection against adverse weather conditions where necessary to maintain their health,” and that “in the case of outdoor access, it is sufficient to offer the animals the possibility of shelter” [26]. This raises the question of what weather conditions (e.g., heat or cold) pose a health risk. Sheep are relatively tolerant to cold, provided their fleece is not wet. This does not apply to newly shorn sheep or newborn lambs, which are wet at birth and have little wool. In hot conditions, it has been observed that sheep prefer sunny spots with airflow, which offers both cooling and protection from insects, over windless shaded areas with high humidity and insect burden.

According to the guidelines of the German Veterinary Association for Animal Welfare, a windbreak facing the prevailing wind direction is considered adequate weather protection. This may consist of closely planted trees or hedges. For ewes with lambs under four weeks old, a covered, three-sided shelter with straw bedding is required if temperatures fall below 0 °C [28]. The fact that requirements differ by veterinary office is illustrated by the guidelines from the StädteRegion Aachen, which extend this age threshold to eight weeks [29].

The German Small Ruminant Veterinary Association also recommends weather protection for freshly shorn sheep in cold and wet weather, as well as in strong sunlight [27]. To prevent exposure to cold, shearing should ideally take place in May or June [27]. A lack of knowledge about appropriate shearing times may contribute to welfare issues, but a shortage of sheep shearers is likely also a factor. Sheep shearing is a rare profession in Germany, and it can be difficult to find someone qualified to perform this task.

However, society must also be better educated regarding the management of small ruminants, especially in regions with a prevalence of transhumance and dike shepherding, to help prevent unfounded reports to veterinary authorities, thereby lessening the burden on both veterinary offices and animal owners. Installing information boards on fencing or hiking trails may be useful for this purpose.

A positive aspect of the survey results is that livestock owners were mostly very understanding when they were informed of existing violations in their sheep husbandry. The relatively low proportion of long-tailed animals with increased diarrhea or maggot problems during inspections is another positive aspects. Other studies have shown that tail docking has no particular effect on the development of myiasis [30,31].

## 5. Conclusions

The method of supra-regional evaluation of officially recorded animal welfare violations is well suited for analyzing focal points in animal husbandry and developing preventive measures based on this knowledge.

## Figures and Tables

**Figure 1 animals-15-02116-f001:**
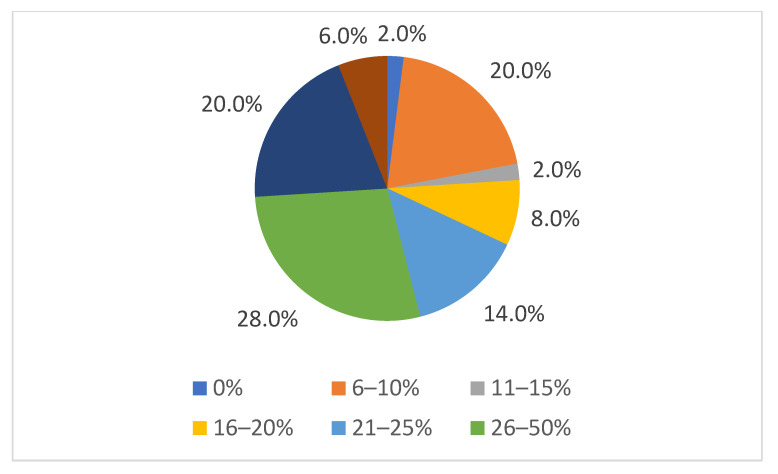
Frequency of repeat cases of animal welfare violations. The percentages at the bottom correspond to the answer options in the questionnaire; the percentages around the pie chart correspond to the percentages of the answer option ticked.

**Figure 2 animals-15-02116-f002:**
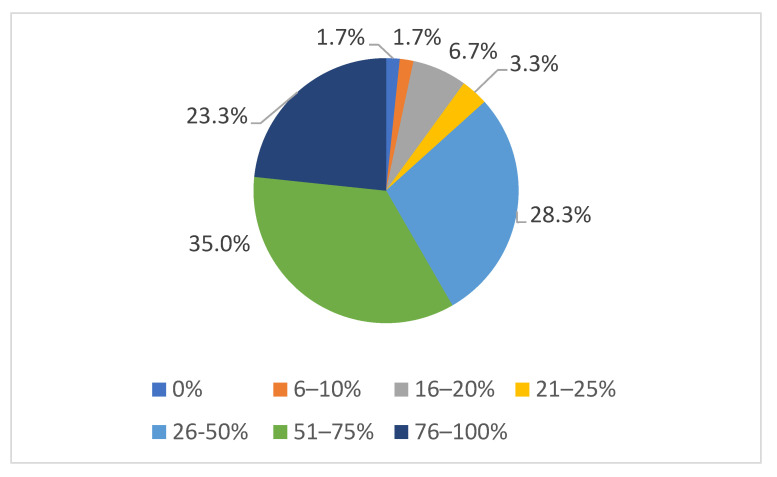
Insight of livestock owners after being made aware of animal welfare violations on their farms. The percentages at the bottom correspond to the answer options in the questionnaire; the percentages around the pie chart correspond to the percentages of the answer option ticked.

**Table 1 animals-15-02116-t001:** Information on the participating veterinary offices by federal state.

Federal States	Number of Participating Veterinary Offices	Percentage of the Veterinary Offices in the Respective Federal State
North Rhine-Westphalia	11	20.4
Bavaria	10	11.5
Baden-Württemberg	7	16.3
Brandenburg	7	38.9
Lower Saxony	7	15.2
Thuringia	6	35.3
Hesse	5	19.2
Rhineland-Palatinate	5	14.7
Schleswig-Holstein	4	28.6
Berlin	3	25.0
Saxony	2	16.7
Saxony-Anhalt	2	15.4
Hamburg	1	14.3
Mecklenburg-Western Pomerania	1	14.3
Saarland	0	0.0
Bremen	0	0.0

**Table 2 animals-15-02116-t002:** Evaluation of the questionnaire in the category “husbandry and housing”.

Problem	n	%
Weather protection	60	90.9
Free-range animals	34	51.5
Stocking density too high	6	9.1
Other	28	42.2

**Table 3 animals-15-02116-t003:** Evaluation of the questionnaire in the category “feed and water supply”.

Problem	n	%
Water supply	63	92.6
Problem with supplementary feeding	38	55.9
Lack of feed in the pasture	32	47.1
No mineral feed	15	22.1

**Table 4 animals-15-02116-t004:** Evaluation of the questionnaire in the category “health”.

Problem	n	%
Lameness	56	83.6
Downer sheep	23	34.3
Skin damage	15	22.4
Diarrhea	13	19.4
High animal losses	13	19.4
Open wounds	11	16.4
Myiasis	5	7.5
Cough/nasal discharge	3	4.5

**Table 5 animals-15-02116-t005:** Evaluation of the questionnaire in the category “lambing”.

Problem	n	%
Increased lamb mortality	56	35.7
Bleating/hungry lambs	11	19.6
Other	9	16.1

**Table 6 animals-15-02116-t006:** Evaluation of the questionnaire in the category “management”.

Problem	n	%
Shearing	52	77.6
Hoof care	40	60.6
Livestock-guarding dogs	8	11.9
Other	6	9.1

**Table 7 animals-15-02116-t007:** Frequency of judicial and extrajudicial settlements in animal welfare cases in sheep farming.

	Judicial		Out of Court	
Frequency	n	%	n	%
0%	19	67.9	67.9	4.8
6–10%	2	7.1	7.1	1.6
11–15%	1	3.6	3.6	0
16–20%	1	3.6	3.6	0
21–25%	3	10.7	10.7	0
26–50%	1	3.6	3.6	1.6
51–75%	0	0	0	19
76–100%	1	3.6	3.6	73

The percentages in the first column correspond to the answer options in the questionnaire; n indicates the number of responses per answer option.

**Table 8 animals-15-02116-t008:** Frequency of animal removals, animal keeping, and animal care bans for animal welfare violations.

	Animal Removal		Animal Husbandry Ban		Ban on Animal Care	
Frequency	n	%	n	%	n	%
0%	20	62.5	24	72.7	27	75.0
6–10%	5	15.6	4	12.1	4	11.1
11–15%	1	3.1	1	3.0	1	2.8
16–20%	1	3.1	0	0	0	0
21–25%	0	0	1	3.0	1	2.8
26–50%	2	6.3	3	9.1	3	8.3
51–75%	3	9.4	0	0	0	0
76–100%	0	0	0	0	0	0

The percentages in the first column correspond to the answer options in the questionnaire; n indicates the number of mentions per answer option.

**Table 9 animals-15-02116-t009:** Frequency of problems during the killing/slaughter of sheep (n = 38).

	Lack of Knowledge/Skills		No Expertise		Killing Without Reasonable Cause		Miscellaneous	
Frequency	n	%	n	%	n	%	n	%
0%	16	42.1	21	55.3	23	60.5	26	68.4
6–10%	1	2.6	0	0	0	0	0	0
11–15%	2	5.3	1	2.6	0	0	0	0
21–25%	0	0	1	2.6	0	0	0	0
26–50%	1	2.6	2	5.3	1	2.6	2	5.3
51–75%	5	13.2	2	5.3	1	2.6	1	2.6
0%	6	15.8	2	5.3	1	2.6	0	0

The percentages in the first column correspond to the answer options in the questionnaire; n indicates the number of mentions per answer option.

**Table 10 animals-15-02116-t010:** Information on publicly accessible court cases by federal state.

Federal States	n	%
North Rhine-Westphalia	6	11.5
Bavaria	16	30.8
Baden-Württemberg	4	7.7
Brandenburg	2	3.8
Lower Saxony	6	11.5
Thuringia	0	0.0
Hesse	4	7.7
Rhineland-Palatinate	6	11.5
Schleswig-Holstein	3	5.8
Berlin	0	0.0
Saxony	1	1.9
Saxony-Anhalt	1	1.9
Hamburg	0	0.0
Mecklenburg-Western Pomerania	3	5.8
Saarland	0	0.0
Bremen	0	0.0

**Table 11 animals-15-02116-t011:** Evaluation of the publicly accessible court cases with regard to the subject matter (n = 52).

Problem	%
Nutrition	50
Water supply	34.6
Weather protection	32.7
Shearing	21.2
Other	21.2
Care	19.2
Parasites	19.2
Omitted treatments	17.3
Fencing	17.3
Stabling	15.4
Hoof care/lameness	11.5
Support	9.6
Labeling	5.8

Several problem areas could be affected in one case.

## Data Availability

The raw data supporting the conclusions of this article will be made available by the authors on request.

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
