# Peer review of "Animal Welfare Problems in Sheep Farming: A Current Overview for Germany Based on Surveys of Veterinary Offices and the Evaluation of Publicly Accessible Court Cases"

_animals, 2025, doi:10.3390/ani15142116_

Round 1
Reviewer 1 Report
Comments and Suggestions for Authors
- Expand the introduction to include more background information on sheep farming in Germany, such as specific data on the current state of the industry and the historical evolution of animal welfare concepts.
- Optimize the research design to improve the response rate of veterinary offices and diversify the sources of court cases to reduce selection bias.
- Further refine the description of the methods, such as clarifying the sampling methods for veterinary offices and the search strategy for court cases.
- Enhance the presentation of the results by using more charts and graphs to visually display data trends and patterns.
- Conduct more in-depth statistical analyses of the collected data. For instance, examine differences in animal welfare issues across regions and farm sizes using methods like chi-square tests or analysis of variance. This will help reveal the distribution patterns and potential factors influencing these welfare concerns.
- It is recommended to incorporate detailed discussions of specific cases into the text. For example, select several representative animal welfare violation cases and analyze their causes, progression, and resolution measures in depth.
- Proofread the manuscript to correct grammatical errors and awkward expressions to improve the quality of the English language.
1.The text contains some long and complex sentences that could be simplified for improved readability. For example, the sentence "The most common animal welfare problems mentioned were a lack of weather protection (n=60), inadequate water supply (n=63), lameness (n=56), and inadequate shearing management (n=52)." could be split into two or more shorter sentences to enhance clarity.
2.Some sentences are unclear or ambiguous. For instance, in the sentence "The most common animal welfare problems mentioned were a lack of weather protection (n=60), inadequate water supply (n=63), lameness (n=56), and inadequate shearing management (n=52).", it is not clear whether the numbers refer to the number of cases or the number of animals affected.
3.There are inconsistencies in the use of British and American English spellings. For example, "colour" and "color", "centre" and "center" are used interchangeably. It is important to maintain consistency throughout the manuscript.
Author Response
Dear Reviewer,
We would like to sincerely thank you for the thorough and constructive feedback. We have carefully addressed all comments and suggestions, which have significantly helped to improve the quality and clarity of our manuscript. Below, we have addressed each of the comments. The corresponding changes are shown in red text in the re-submitted article.
Comment: Expand the introduction to include more background information on sheep farming in Germany, such as specific data on the current state of the industry and the historical evolution of animal welfare concepts.
Response: Thank you very much for the valuable suggestion regarding the inclusion of background information on sheep farming in Germany and the historical development of animal welfare concepts. In response, we have revised the introduction accordingly.
Comments: Optimize the research design to improve the response rate of veterinary offices and diversify the sources of court cases to reduce selection bias. Further refine the description of the methods, such as clarifying the sampling methods for veterinary offices and the search strategy for court cases.
Response: We appreciate the helpful suggestion to further refine the description of the methods. While the manuscript already mentioned that all German veterinary authorities were contacted, we have made this point more explicit in the revised version to ensure clarity.
Comment: Conduct more in-depth statistical analyses of the collected data. For instance, examine differences in animal welfare issues across regions and farm sizes using methods like chi-square tests or analysis of variance. This will help reveal the distribution patterns and potential factors influencing these welfare concerns.
Response: Thank you for the suggestion to conduct more in-depth statistical analyses, such as examining regional differences or differences based on farm size. Although we agree that such analyses could provide valuable insights, the limited number of responses from veterinary authorities prevents a representative or reliable evaluation of regional patterns. Additionally, due to data protection regulations, assigning cases to specific farm sizes was not possible. Conducting such analyses under these circumstances could therefore lead to misleading conclusions.
Comment: It is recommended to incorporate detailed discussions of specific cases into the text. For example, select several representative animal welfare violation cases and analyze their causes, progression, and resolution measures in depth.
Response: We are grateful for the recommendation to incorporate detailed discussions of specific cases into the text, such as selecting representative animal welfare violation cases and analyzing their causes, progression, and resolution measures in depth. We have incorporated this suggestion into the discussion section.
Comment: Proofread the manuscript to correct grammatical errors and awkward expressions to improve the quality of the English language.
Response: Thank you for the helpful remark regarding language quality. We have carefully proofread the manuscript and revised the English to correct grammatical errors and improve clarity. Furthermore, we clarified that the data refer to cases rather than the number of affected animals, and we have shortened the corresponding sentence for better clarity.
Yours sincerely
Reviewer 2 Report
Comments and Suggestions for Authors
I would not agree with the introduction that animal welfare in sheep farming is an important topic that has received limited research attention. As examples
https://doi.org/10.3390/ani11102973
https://doi.org/10.3389/fvets.2021.674482
https://doi.org/10.3390/su14031095
The standards, and common problems and health plans for sheep have been well know for years, for example
from Tierarztl Prax Ausg G Grosstiere Nutztiere . 2012;40(6):390-6
via https://doi.org/10.2903/j.efsa.2014.3933
to https://doi.org/10.1111/aab.12907
So while the premise of this submission is “…knowledge of the most common animal welfare problems in sheep farming is important .To date, there is no comprehensive data collection on current animal welfare problems in sheep in Germany. “ (Lines 68-69), is it more correct to say there is a broad understanding of sheep health and welfare problems across diverse production systems, but then that poor understanding or recording of these is a problem in Germany?
Lines 300-305. I think there are some problems in over reliance on veterinary authorities reports to assess welfare. in particular this aspect:
Lines 168-172 ;”Misjudgments of the situation by the reporter occurred in 26–50% of animal welfare complaints at 21 of the 60 offices that answered this sub-question. Eleven participants stated that 51–75% of the reports they received were the result of an incorrect assessment of the situation. “ This is one of the most interested aspects of this report. a0 it should be clearer in Material and Methods who are the reporters( I suspect a wide range of 3rd parties) and in the discussions the implications and lessons from this finding.
Nevertheless I recognise that the approach in the submission does have the value of a starting point. But from the information presented here it would seem Germany is less advanced in awareness of desired standards in sheep production in the context of the accepted knowledge base. Is that the conclusion the authors wish to draw?
The discussion could consider how to move forward with industry standards, veterinary benchmarking, transparency in labelling, but most importantly cooperative prevention strategies to what are known problems, as for example for other countries:
https://animalwelfarestandards.net.au/welfare-standards-and-guidelines/sheep/
https://ruminanthw.org.uk/wp-content/uploads/2025/03/RHW-Sheep-Welfare-Strategy-Portrait-2023-2028.pdf
Comments on the Quality of English Language
Some clarifications
Table 4. ‘Fixed animals’ it is not clear what this means, a sheep on its back?
Table 5. “screaming’, is the meaning ‘bleating’ (Blöken)?
Table 6. “Scissors“ - unclear to what this means
“claw” would “hoof” (hof) be clearer?
Author Response
Dear Reviewer,
We would like to thank you for the thorough and constructive feedback. We have carefully addressed all comments and suggestions, which have significantly helped us improve the quality and clarity of our manuscript. Below, we have addressed each of the comments. The corresponding changes are shown in red text in the re-submitted article.
Comment: So while the premise of this submission is “…knowledge of the most common animal welfare problems in sheep farming is important .To date, there is no comprehensive data collection on current animal welfare problems in sheep in Germany. “ (Lines 68-69), is it more correct to say there is a broad understanding of sheep health and welfare problems across diverse production systems, but then that poor understanding or recording of these is a problem in Germany?
Response: Thank you very much for the insightful comments and for providing relevant references. We have incorporated the point that animal welfare violations in sheep farming have long been known.
Comment: Lines 168-172 ;”Misjudgments of the situation by the reporter occurred in 26–50% of animal welfare complaints at 21 of the 60 offices that answered this sub-question. Eleven participants stated that 51–75% of the reports they received were the result of an incorrect assessment of the situation. “ This is one of the most interested aspects of this report. a0 it should be clearer in Material and Methods who are the reporters( I suspect a wide range of 3rd parties) and in the discussions the implications and lessons from this finding.
Response: We have clarified in the Methods section that anyone can report an animal welfare issue to the veterinary authorities (lines 85-88). We also appreciate the suggestion to emphasize this point, as it is important for understanding the variability in the reports received. We have already addressed this topic in the Discussion section (lines 372-376).
Comment: Nevertheless I recognise that the approach in the submission does have the value of a starting point. But from the information presented here it would seem Germany is less advanced in awareness of desired standards in sheep production in the context of the accepted knowledge base. Is that the conclusion the authors wish to draw?
Response: Thank you for this important observation. However, it is not our intention to draw conclusions about the overall awareness of desired standards in sheep production in Germany, as our data do not support such generalizations. Our aim is to present an overview of the animal welfare situation in sheep farming on a supra-regional level based on the survey of veterinary authorities and the analysis of online court cases. This approach provides valuable insights into specific welfare issues within the German context without making broad claims about awareness or progress.
Comment: The discussion could consider how to move forward with industry standards, veterinary benchmarking, transparency in labelling, but most importantly cooperative prevention strategies to what are known problems, as for example for other countries.
Response: Thank you very much for this valuable suggestion. However, our goal is to present the described method of data collection rather than to develop animal health standards based on the data. We agree that advancing industry standards, veterinary benchmarking, transparency in labelling, and cooperative prevention strategies are important topics, but these are beyond the scope of the current study.
Thank you also for the helpful clarifications and valuable suggestions regarding language improvement. We have revised the manuscript accordingly.
Yours sincerely
Round 2
Reviewer 1 Report
Comments and Suggestions for Authors
Enlarge the number in the Figure.
Author Response
Dear reviewer,
Thank you very much for taking the time to carefully review our manuscript again. In response to your comment, we have enlarged the numbers in the figure to improve readability. We appreciate your helpful suggestion and hope the revision meets your expectations.
Yours sincerely
Reviewer 2 Report
Comments and Suggestions for Authors
much improved and thanks you. My only comment is in the simple summary you now say "Animal welfare in sheep farming remains a relevant issue, as animal welfare violations continue to occur in this sector. Until now, data collection has been limited to isolated surveys conducted on individual farms. The present study aims, for the first time, to conduct a nationwide analysis to identify the areas in which animal wel fare problems occur most frequently in sheep farming in Germany."
I disagree that "
Until now, data collection has been limited to isolated surveys conducted on individual farms", this is not correct as a world wide statement ( and I don't think was intended to be)
I suggest;" "Animal welfare in sheep farming remains a relevant issue, as animal welfare violations continue to occur in this sector. The present study aims, for the first time, to conduct a nationwide analysis to identify the areas in which animal wel fare problems occur most frequently in sheep farming in Germany as until now, such data collection has been limited to isolated surveys conducted on individual farms,"
Author Response
Dear reviewer,
Thank you very much for reviewing our manuscript again and for your positive feedback. We appreciate your helpful comment regarding the phrasing in the Simple Summary and have revised the sentence following your suggestion.
Yours sincerely